# Effects of Seed Bio-Priming by Purple Non-Sulfur Bacteria (PNSB) on the Root Development of Rice

**DOI:** 10.3390/microorganisms10112197

**Published:** 2022-11-06

**Authors:** Ranko Iwai, Shunta Uchida, Sayaka Yamaguchi, Fumika Sonoda, Kana Tsunoda, Hiroto Nagata, Daiki Nagata, Aoi Koga, Midori Goto, Taka-aki Maki, Shuhei Hayashi, Shinjiro Yamamoto, Hitoshi Miyasaka

**Affiliations:** 1Department of Applied Life Science, Sojo University, 4-22-1 Ikeda, Nishiku, Kumamoto 860-0082, Japan; 2Ciamo Co., Ltd., Sojo University, 4-22-1 Ikeda, Nishiku, Kumamoto 860-0082, Japan; 3Matsumoto Institute of Microorganisms Co., Ltd., 2904 Niimura, Matsumoto, Nagano 390-1241, Japan

**Keywords:** purple non-sulfur bacteria, *Rhodopseudomonas* sp., *Rhodobacter* sp., rice, bio-priming, lipopolysaccharide (LPS)

## Abstract

The effects of seed bio-priming (seed soaking) with purple non-sulfur bacteria (PNSB) on the grain productivity and root development of rice were examined by a field study and laboratory experiments, respectively. Two PNSB strains, *Rhodopseudomonas* sp. Tsuru2 and *Rhodobacter* sp. Tsuru3, isolated from the paddy field of the study site were used for seed bio-priming. For seed bio-priming in the field study, the rice seeds were soaked for 1 day in water containing a 1 × 10^5^ colony forming unit (cfu)/mL of PNSB cells, and the rice grain productivities at the harvest time were 420, 462 and 504 kg/are for the control, strain Tsuru2-primed, and strain Tsuru3-primed seeds, respectively. The effects of seed priming on the root development were examined with cell pot cultivation experiments for 2 weeks. The total root length, root surface area, number of tips and forks were evaluated with WinRhizo, an image analysis system, and strains Tsuru2- and Tsuru3-primed seeds showed better root development than the control seeds. The effects of seed priming with the dead (killed) PNSB cells were also examined, and the seed priming with the dead cells was also effective, indicating that the effects were attributed to some cellular components. We expected the lipopolysaccharide (LPS) of PNSB as the effective component of PNSB and found that seed priming with LPS of *Rhodobacter sphaeroides* NBRC 12203 (type culture) at the concentrations of 5 ng/mL and 50 ng/mL enhanced the root development.

## 1. Introduction

Purple non-sulfur bacteria (PNSB) are phototrophic microorganisms. They have photosynthetic reaction centers and light-absorbing pigments and convert light energy into chemical energy (ATP). Unlike cyanobacteria, they do not use water as an electron donor for photochemical reactions, and they do not emit oxygen. This type of photosynthesis is called anoxygenic photosynthesis. PNSB have a wide variety of biotechnological applications in agriculture [1,2,3,4], aquaculture [5,6], biomaterial production [7,8,9], renewable energy production [10,11], and bioremediation [12]. Among these applications, one of the most widely applied practices is the plant growth promotion by PNSB in agriculture. The mechanisms of plant growth promotion by PNSB have been intensively studied [1,3], and the mechanisms can be classified into (1) the biological activities of PNSB and (2) the production of various beneficial (plant growth promoting) substances by PNSB. The biological activities include nitrogen fixation, phosphate solubilization and hydrogen sulfide degradation. The production of beneficial substances includes the production of polyphosphate, pigments, vitamins and plant growth-promoting substances (PGPSs), such as indole-3-acetic acid (IAA) and 5-aminolevulinic acid (ALA). In addition, we recently proposed lipopolysaccharide (LPS), a cell wall component of gram-negative bacteria, as a new active principle of PNSB [13].

Seed priming is a pre-sowing treatment to improve germination, and also overall plant growth and/or yield [14]. Seed bio-priming is a process that treats seeds with beneficial biological agents such as phytohormones and microbes [15]. Seed bio-priming with plant growth-promoting microbes is one of the most promising approaches, and its effects are the promotion of germination, the promotion of growth after germination, the induction of resistance against biotic and abiotic stresses, etc. The strains mainly used for bio-priming belong to *Trichoderma* spp., *Enterobacter* spp., *Pseudomonas* spp. and *Bacillus* spp. [16]. In Japan, seed bio-priming by PNSB has been practiced among some rice farmers to improve the growth of rice seedlings and to increase the rice grain yield, but there have been no studies that evaluated the effects of this method quantitatively. In this study, we investigated the effects of seed bio-priming by PNSB by conducting a field study in a commercially operating rice farm and laboratory experiments.

## 2. Materials and Methods

### 2.1. Rice Cultivars

Two rice (*Oryza sativa*) cultivars, Kumasan-no-chikara and Akita-komachi, were used. Kumasan-no-chikara is an original rice cultivar of Kumamoto prefecture, Japan. Akita-komachi is one of the most popular cultivars in Japan and it is also cultivated among several East Asian countries. The seeds of Kumasan-no-chikara and Akita-komachi were purchased from a local branch of Japan Agricultural Cooperatives (JA) and Noken Co., Ltd. (Kyoto, Japan), respectively.

### 2.2. Bacterial Strains and Culture Condition

Three PNSB strains, *Rhodopseudomonas* sp. Tsuru2, *Rhodobacter* sp. Tsuru3, and *Rhodobacter sphaeroides* NBRC 12203^T^ (or ATCC 17023) were used. The strains Tsuru2 and Tsuru3 were isolated from the soil of a paddy field in the study site (Sakamoto-cho Tsurubami, Yatsushiro, Kumamoto, Japan), and these strains grew well in the medium made using the distillation remnants of Kuma Shochu, a Japanese local spirit, without any additives [17]. These strains were identified without purifying the culture by amplifying the *pufL* and *pufM* (*pufLM*) region of the *puf* operon, which codes for the subunits of the light-harvesting and the reaction center complexes of PNSB [18,19], as described previously [17]. The GenBank/EMBL/DDBJ accession numbers of the *pufLM* genes of strains Tsuru2 and Tsuru3 are LC617215 and LC617217, respectively. *R. sphaeroides* NBRC 12203 ^T^, the type culture of *Rhodobacter sphaeroides*, was obtained from the NBRC (Biological Resource Center, NITE, Japan) culture collection.

The strains Tsuru2 and Truru3 were cultured by using a Kuma-Red^TM^ PNSB culture kit (Ciamo Co., Ltd., Kumamoto, Japan). Briefly, 20 mL of concentrated culture broth made from distillation remnants of Kuma Shochu was dissolved in 1.9 L of tap water in a 2 L size PET bottle, and 50 mL of seed culture (approximate concentration of 10^9^ cuf/mL) was added to the bottle. The bottle was placed horizontally in light conditions. *R. sphaeroides* NBRC 12203^T^ was cultured in glutamate–malate medium [20]. PNSB were cultured at 25 °C in light conditions (light intensity ca. 100 mmol photon/s/m^2^).

For the preparation of dead (killed) PNSB cells, the cells were collected by centrifugation, and the cell pellet was dried at 60 °C for 6 h. Distilled water was added to the dried pellet, and the pellet was disrupted by sonication.

### 2.3. LPS from Rhodobacter sphaeroides NBRC 12203 ^T^

LPS of *R. sphaeroides* ATCC 17023 [21] was purchased from InvivoGen (San Diego, CA, USA). *R. sphaeroides* NBRC 12203^T^ and *R. sphaeroides* ATCC 17023 are the identical strains with different culture collection numbers. To avoid confusion, in the present paper we call this LPS as LPS from *R. sphaeroides* NBRC 12203^T^.

### 2.4. Design of Field Study

The field study to examine the effects of bio-priming by PSNB on the rice grain yield was performed with the help of a local agricultural producers’ cooperative corporation, Tsurubami Nanohana-mura, Yatsushiro, Kumamoto, Japan. This corporation uses 6.42 ha of field for rice cultivation, and they divided the field into 3 parts: 1.96 ha for control (no bio-priming), 2.05 ha for strain Tsuru2 bio-priming, and 2.41 ha for strain Tsuru3 bio-priming. Bio-priming by PNSB was completed for one day at the cell concentration of 10^5^ colony forming unit (cfu)/mL. The procedure for bio-priming by PNSB is shown in Appendix A. The study was performed from April to October in 2020.

### 2.5. Bio-Priming of Rice Seeds by PNSB and LPS (Laboratory Experiments)

Rice seeds were soaked in water at 25 °C for 24 h, and then soaked in PNSB or LPS solution at 25 °C for 24 h. The control seeds were soaked in distilled water in the same way. After bio-priming, the rice seeds were sown in a cell pot (49 cells, cell volume: 30 cm^3^) containing PROMIX BX soil (Premier Horticulture Ltd., Richer, MB, Canada), one seed per pot. The cell pots were irrigated daily and kept at 25 °C under 12 h of light conditions (light intensity ca. 100 mmol photon/s/m^2^) and 12 h of dark conditions.

### 2.6. Analysis of Root Development by WinRhizo Image Analyzing System

The images of the roots were obtained with a scanner (Epson GT-X970, Seiko Epson Co., Ltd., Suwa, Japan) and they were analyzed by automatic image processing with WinRhizo software (Regent Instruments Inc., Québec, QC, Canada). Data on total root length, root surface area, root tips, root forks were obtained. The roots were partitioned into 10 diameter classes in 0.05 mm steps (0 mm–0.45 mm) and root lengths for each root diameter class were also obtained. The percentage of thin roots (less than 0.1 mm diameter) was calculated as {(length of root of less than 0.1 mm diameter)/(total root length)} × 100.

### 2.7. Statistical Methods

For statistical analyses, student’s t-test was used for pairwise comparison, and one-way ANOVA with a post-hoc Duncan’s test was used for multiple comparison.

## 3. Results

### 3.1. Effects of PNSB Bio-Priming on Rice Yield (Field Study)

For rice farming in Japan, rice seeds are first sown onto a tray for germination and seedling growth, and about one month after sowing, the seedlings on the tray are sent to a special planting machine (vehicle) to plant the seedlings to flooded paddy field (see Appendix A). Bio-priming by PNSB was completed at the cell concentration of 10^5^ colony forming unit (cfu)/mL for one day before sowing the seeds onto the tray. Figure 1 shows the comparison of the root development of the control (Figure 1a), the strain Tsuru2-primed (Figure 1b) and strain Tsuru3-primed (Figure 1c) seedlings before planting in the paddy field. The roots of the PNSB-primed seedling showed a better development of roots, and in particular the generation of a lot of white-colored, thin lateral roots was observed.

Table 1 shows the yields of rice gains (brown rice) at the harvest time in September. The yields of rice gains were 420, 462 and 504 kg/are for the control, strain Tsuru2-primed, and strain Tsuru3-primed seeds, respectively.

### 3.2. Effects of Bio-Priming by Strain Tsuru2 and Strain Tsuru3 on the Root Development of Rice (Laboratory Experiments)

The field study suggested that the seed bio-priming by PNSB promoted the root development after germination, and the well-developed roots support the further growth of rice, resulting in the higher yield in rice grain. To quantitatively evaluate the effects of bio-priming by PNSB on root development, laboratory experiments using 7 × 7 cell pot experiments were performed. The same rice cultivar, Kumasan-no chikara, and the same PNSB strains, strain Tsuru2 and strain Tsuru3 (at the cell concentration of 10^5^ cfu/mL) used for the field study, were used for the laboratory experiments. To examine whether the effects of bio-priming were attributed to the biological activity of the live PNSB cells or whether some cellular components of PNSB were acting as active principles, the bio-priming by both live and dead (killed) PNSB cells was examined. The seedling were grown for 14 days, and the root development was evaluated by a WinRhizo image analyzing system. Figure 2 shows the results. For the live cells of strain Tsuru3, the root length, surface area, and number of tips and forks were significantly increased, while for the dead cells of strain Tsuru3, the root length and number of tips and forks were significantly increased. For the live and dead cells of strain Tsuru2, the values of root length, surface area, and number of tips and forks were higher than those of the control, but the differences were not significant. For strain Tsuru3, the effect of the dead cells was smaller than that of the live cells, while for strain Tsuru2, there was no difference between the effects of live and dead cells. The field study suggested that the bio-priming by PNSB promoted the generation of thin lateral roots (Figure 1), and we also calculated the percentage of thin (less than 0.1 mm root diameter) roots (Figure 2e). As expected, the percentages of thin roots were significantly higher in all the treatments, indicating that the bio-priming by PNSB promoted the generation of thin lateral roots.

### 3.3. Effects of Bio-Priming by R. Sphaeroides NBRC 12203^T^ and its LPS on the Root Development of Rice

The field and laboratory studies with strains Tsuru2 and Tsuru3 indicated the promotive effects of seed bio-priming by PNSB on the root development. The strains Tsuru2 and Tsuru3 were the study-site-original and practically applied PNSB strains. These strains were, however, not pure cultures, because they were cultured by using a Kuma-Red^TM^ PNSB culture kit (Ciamo Co., Ltd., Kumamoto, Japan) which was designed to culture PNSB onsite by farmers [17]. We therefore could not exclude the possibility of the influences of the coexisting microorganisms in the PNSB culture solution. To confirm that the effects of bio-priming are attributed to PNSB, we examined the effects of seed bio-priming with a pure culture of *R. sphaeroides* NBRC 12203^T^. A rice cultivar, Akita-komachi, which is one of the most widely used Japanese cultivars, was used, and the seeds were primed with *R. sphaeroides* NBRC 12203^T^ (live cell) at the concentration of 1 × 10^5^ cfu/mL (the same concentration as for strains Tsuru2 and Tsuru3). Figure 3 shows the results. The root development (root length and number of tips) was significantly enhanced, and the generation of thin roots was also significantly increased by bio-priming by *R. sphaeroides* NBRC 12203^T^. The values of the surface area and number of forks were also higher than those of the control, but the differences were not significant. We also examined the effects of the dead cells of *R. sphaeroides* NBRC 12203^T^, and the dead cells also showed root development promotion (see Appendix A). However, a significant difference was observed only for the root length, and the other values (surface area, number of tips and forks, and % of thin roots) were higher than those of the control but not significant.

In the previous study [13], we proposed LPS as a new active principle of PNSB. We expected that LPS could possibly be acting as an active principle of seed bio-priming by PNSB because the dead (killed) cells also showed the same effects as shown Figure 2, and we examined the effect of seed bio-priming by LPS of *R. sphaeroides* NBRC 12203^T^ on root development. The effective concentration of PNSB culture for bio-priming was 1 × 10^5^ cfu/mL, and this value is equal to about 500 ng cell dry weight/mL. The LPS content in bacterial cells is reportedly about 5 to 10% of total cell dry weight, and based on this information, we examined the effect of LPS at the concentrations of 5 ng/mL and 50 ng/mL. Figure 4 shows the results. Significant differences were observed in the values of root length (5 ng/mL and 50 ng/mL), surface area (5 ng/mL) and the number of forks (5 ng/mL and 50 ng/mL). The number of tips and % of thin roots were higher than those of the control, but the differences were not significant.

## 4. Discussion

PNSB have been widely applied as eco-friendly agricultural biomaterials all over the world [1,3]. For example, in Japan, PNSB are applied to plants by foliar spray and/or land irrigation for vegetables and fruits, and by pouring them into flooded paddy fields and lotus ponds [17,22]. In addition, some rice farmers in Japan use PNSB for the bio-priming of rice seeds to enhance the root development and increase the grain yield at the harvest time. There have been many studies on bio-priming with plant growth-promoting microbes, such as *Agrobacterium* spp., *Arthrobacter* spp., *Azotobacter* spp., *Azospirillum* spp., *Bacillus* spp., *Enterobacter* spp., *Burkholderia* spp., *Klebsiella* spp., *Pseudomonas* spp., *Rhizobium* spp., *Streptomyces* spp., and *Trichoderma* spp. [16,23], but as far as we know there have been no studies on bio-priming by PNSB. In this study, we first evaluated the effects of bio-priming by PNSB of rice seeds through a field study and laboratory experiments.

In our field study, the yields of rice grain were increased by PNSB bio-priming, indicating the effectiveness of this method. The field study of the present study was, however, performed with commercially operating paddy fields, and due to this limitation, the conditions of the three paddy fields for the control, strain Tsuru2-primed and strain Tsuru3-primed, were not strictly equal. Therefore, further experimental studies under strictly controlled conditions, such as pot cultivation experiments until rice grain harvest, are required in the future.

Our laboratory experiments with strains Tsuru2 and Tsuru3 indicated the promotive effects of PNSB bio-priming on root development. The increase of the percentage of thin roots by PNSB bio-priming (Figure 2) suggested that it also promotes the development of lateral root and root hair formation, and the promotion of lateral root development was also observed in the field study (Figure 1). Lateral roots are derived from a parental root and have a predominant role in the uptake of nutrients and water [24], and therefore the promotion of lateral roots generation by PNSB bio-priming may have enhanced the nutrient uptake by plants. The promotion of lateral root formation by bio-priming with growth-promoting rhizobacteria (PGPR), including *Pseudomonas fluorescens* [25], *Enterobacter hormaechei* [26], *Serratia quinivorans* [27], *Azospirillum lipoferum* CRT1 [28], and *Trichoderma* spp. [29], has also been reported.

Since the strains Tsuru2 and Tsuru3 were not pure PNSB cultures, we also examined the effect of bio-priming by pure culture of *R. sphaeroides* NBRC 12203^T^ and confirmed that the effect was attributed to PNSB.

Both the bio-priming by live and dead PNSB cells showed promotive effects on root development, indicating that some cellular components may be acting as the active principle of PNSB bio-priming, and we showed that the LPS of PNSB was acting at least as one of the active principles of bio-priming (Figure 4). In our previous study [13], we proposed LPS as the active principle of the plant growth promoting effects of PNSB. LPS is the cell wall component of gram-negative bacteria, and it is also known as an endotoxin which causes inflammatory responses at a quite low concentration (pg/mL to ng/mL level) through the Toll-like receptor 4 (TLR4) signaling pathway in animals [30]. In plants, the receptor for LPS, named lipooligosaccharide-specific reduced elicitation (LORE), has been identified, but compared to animals, plants are believed to be much less sensitive to LPS [31]. There have been several reports on the effects of LPSs on plant cells and tissues, and these effects include reactive oxygen species (ROS) generation [32,33,34], induction of secondary metabolites production [35], induction of resistance to plant pathogens, such as fungi [36] and nematode [37], and growth promotion [38]. In all the previous studies reporting the effects of LPSs from various bacteria on plants, however, the effective LPS concentration was 10 to 100 μg/mL (for more details see Appendix A), which are a thousand to a million times higher than the effective concentration range (pg/mL to ng/mL level) in animals. For example, Vallejo-Ochoa et al. reported the ROS generation and growth promotion by LPS from a rhizobacteria, *Azospirillum brasilense* Sp245, in wheat (*Triticum aestivum* L.), but the effective concentration was 100 μg/mL [38]. The bio-priming by LPS of *R. sphaeroides* showed promotive effects on the root development at concentrations of as low as 5 ng/mL to 50 ng/mL (Figure 4), and as we reported in the previous study [13], when the plants (*Brassica rapa* var. *perviridis*) were treated with LPS of *R. sphaeroides* by foliar feeding, the effective concentration was 10 to 100 pg/mL. The very low effective dosage of LPS from *R. sphaeroides* in plants indicated a quite unique biological property of LPS from PNSB among the LPSs from various gram-negative bacteria. Interestingly, in animal cells, LPS from *R. sphaeroides* NBRC 12203^T^ lacks endotoxic activity and acts as an antagonist against toxic LPSs (endotoxins) from various gram-negative bacteria in the TLR4 signaling pathway [39]. The mechanism of antagonistic activity of LPS from *R. sphaeroides* has also been studied, and the unique biological property can be explained by its unique structure of lipid A [40]. Lipid A, a domain of LPS, is a hydrophobic molecule that anchors LPS to the outer membrane of gram-negative bacteria [41], and the biologically active component of LPSs. The lipid A of *R. sphaeroides* showed antagonistic activity for TLR4, the chemically synthesized lipid A of *R. sphaeroides*, named eritoran (E5564; Eisai), has also been developed for therapeutic application [42], and eritoran has been shown to protect animals from lethal cytokine storm by blocking the TLR4 signaling pathway [43]. These previous studies in animal cells suggest that the unique property of LPS of *R. sphaeroides* NBRC 12203^T^ with respect to sensitivity in plant cells could possibly be attributed to the structure of lipid A, and detailed studies on the relationship between lipid A structure and the susceptibility of LPSs in plants are required in the future.

From the viewpoint of practical application in agriculture, bio-priming of rice seeds by PNSB is a quite eco-friendly and cost-effective method. The effective concentration for bio-priming found in this study was as low as 10^5^ cfu/mL, which is ten thousand times the dilution of the original culture of PNSB (10^9^ cfu/mL). Our results indicated the effect of bio-priming by PNSB was higher in live cells than that of dead cells (Figure 2), but for the practical applications dead cells may have a lot of advantages compared to live cells, such as easier storage and handling, stable quality, a lower barrier for international trading, etc., thus the bio-priming by dead PNSB cells is also a promising approach, and further improvement of the dead cell method is under progress in our laboratory.

## 5. Conclusions

Seed bio-priming with plant growth-promoting microbes is one of the most promising approaches to improve plant growth after germination. In this study, we showed for the first time the effectiveness of bio-priming by PNSB. The effective concentration for bio-priming found in this study was as low as 10^5^ cfu/mL, which was ten thousand times the dilution of the original culture of PNSB (10^9^ cfu/mL), thus bio-priming by PNSB is a quite eco-friendly and cost-effective approach to enhancing plant growth. Bio-priming by dead PNSB cells was also effective, and we showed that LPS was acting as the active principles of PNSB bio-priming. In the previous study, we first proposed LPS for the active principle of the plant-growth-promoting effect of PNSB, and the results of the present study provided additional evidence for our proposal. Bio-priming by LPS of *R. sphaeroides* showed promotive effects on root development at concentrations of as low as 5 ng/mL to 50 ng/mL, which were about a thousand times lower than those in the previous studies on the effects of LPSs from various gram-negative bacteria on plants, highlighting the unique biological features of LPS from *R. sphaeroides*.

## Figures and Tables

**Figure 1 microorganisms-10-02197-f001:**
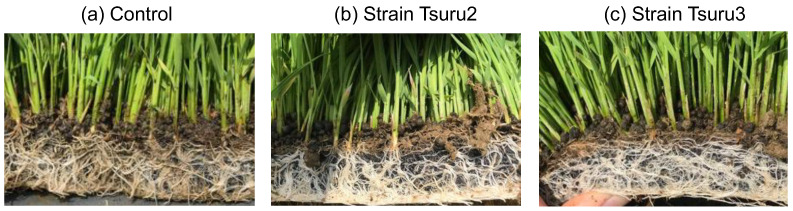
The roots of the rice seedlings before planting in the paddy field (1 month after sowing). (**a**) Control without bio-priming, (**b**) bio-primed with strain Tsuru2, (**c**) bio-primed with strain Tsuru3.

**Figure 2 microorganisms-10-02197-f002:**
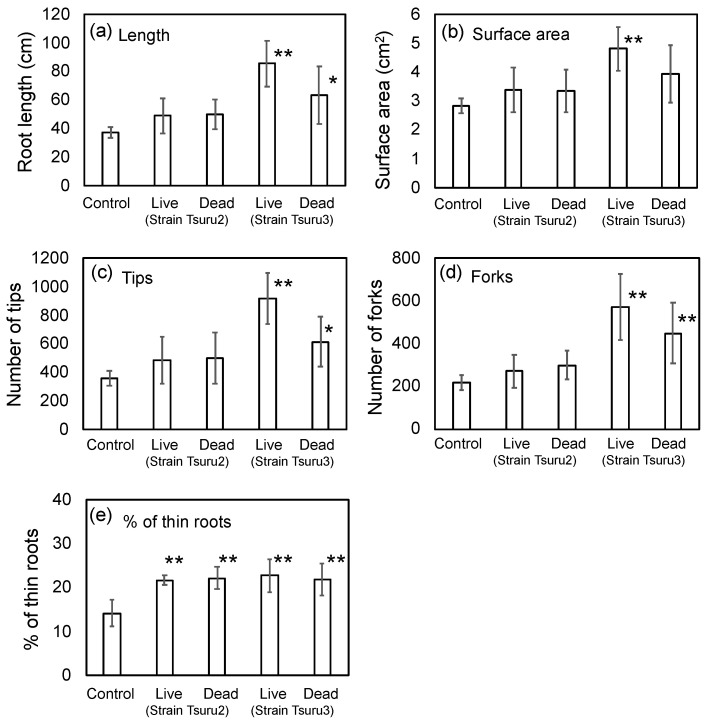
Effect of bio-priming by strains Tsuru2 (live and dead) and Tsuru3 (live and dead) on (**a**) root length, (**b**) surface area, (**c**) number of tips, (**d**) number of forks, and (**e**) % of thin (0.1 mm < diameter) roots. The rice seeds were bio-primed by PNSB for 24 h, sown onto a cell pot and cultivated for 14 days. Data are means ± SD (*n* = 5). An asterisk indicates the result that was significantly different (* *p* < 0.05, ** *p* < 0.01) from the control.

**Figure 3 microorganisms-10-02197-f003:**
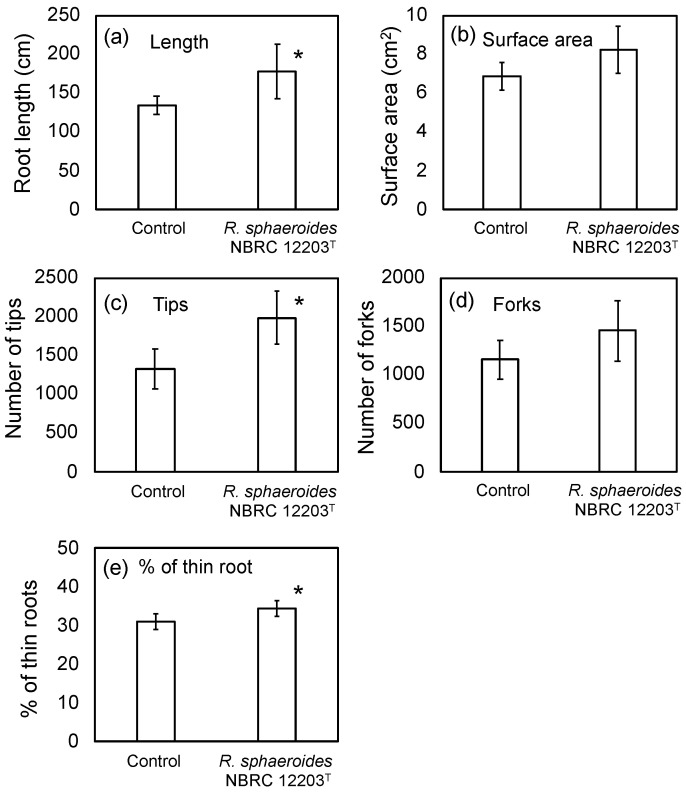
Effect of bio-priming by *R. sphaeroides* NBRC 12203^T^ (live cell) on (**a**) root length, (**b**) surface area, (**c**) number of tips, (**d**) number of forks, and (**e**) % of thin (0.1 mm < diameter) roots. The rice seeds were bio-primed by PNSB for 24 h, sown onto a cell pot and cultivated for 14 days. Data are means ± SD (*n* = 5). An asterisk indicates the result that was significantly different (* *p* < 0.05) from the control.

**Figure 4 microorganisms-10-02197-f004:**
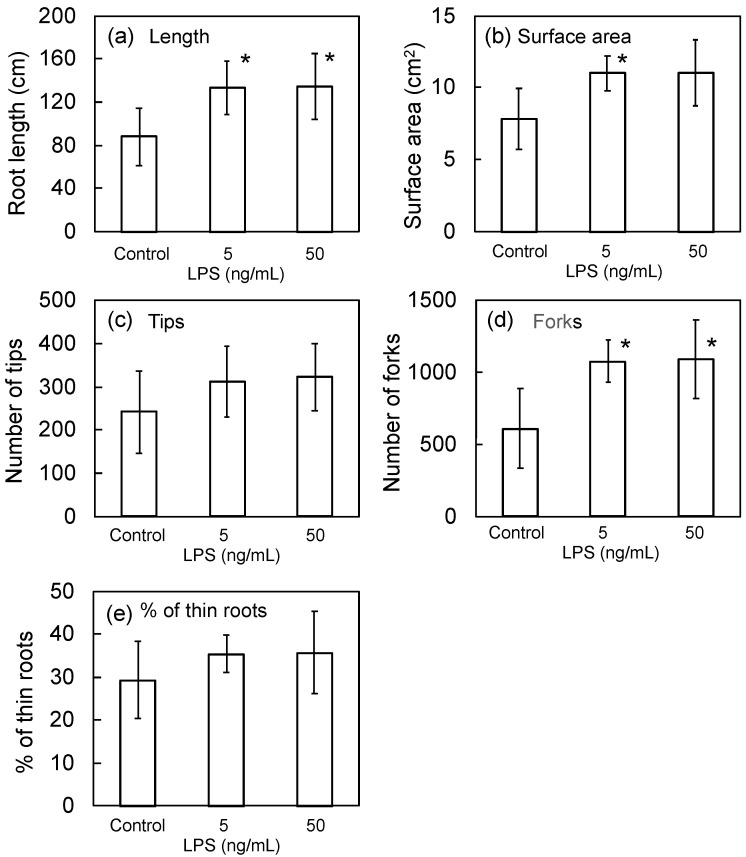
Effect of bio-priming by LPS of *R. sphaeroides* NBRC 12203^T^ on (**a**) root length, (**b**) surface area, (**c**) number of tips, (**d**) number of forks, and (**e**) % of thin (0.1 mm < diameter) roots. The rice seeds were bio-primed by LPS solution for 24 h, sown onto a cell pot and cultivated for 14 days. Data are means ± SD (*n* = 5). An asterisk indicates the result that was significantly different (* *p* < 0.05) from the control.

**Table 1 microorganisms-10-02197-t001:** Yields of rice gains from the control, strain Tsuru2-primed and strain Tsuru3-primed paddy field.

	Yields of Rice Gains (kg/are) ^1^
Control	420
Strain Tsuru2-primed	462
Strain Tsuru3-primed	504

^1^ Average values from the paddy fields of 1.96 ha of control (no bio-priming), 2.05 ha of strain Tsuru2 bio-primed, and 2.41 ha of strain Tsuru3 bio-primed.

## Data Availability

The data presented in this study are available upon request from the corresponding author.

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
