# Peer review of "Effects of Seed Bio-Priming by Purple Non-Sulfur Bacteria (PNSB) on the Root Development of Rice"

_microorganisms, 2022, doi:10.3390/microorganisms10112197_

Round 1

Reviewer 1 Report

This manuscript has described improvement of rice grain productivity by soaking of rice seeds in cell suspension of purple nonsulfur bacteria isolated from regional paddy fields through their effectiveness on rice root development. The enhancement of root development was observed in case of soaking the seeds in solution of LPS collected from type culture strain of Rhodobacter sphaeroides. The conclusions obtained from this study are new and the soaking of rice seeds in the bacterial suspension is practical for farmers. However, as crucial information to evaluate logical accuracy and to reproduce the experiments is lacking, the authors need to supplement the information described below and need to correct inadequate points listed below.

A method to prepare LPS from R. sphaeroides NBRC12203 should be included in the manuscript. In addition, describe how to confirm the identity and purity of LPS in the fraction and how to measure the quantity of LPS.

The cultivation method of purple nonsulfur bacteria (Tsuru2, Tsuru3, and NBRC12203) is not described in the manuscript. Even though readers would look at reference 17 to know a method used for cultivation of Tsuru2 and Tsuru3, ref 17 includes a method using 24 well plate. I think a larger culture volume might be required for the rice seed soaking. The information for ref 17 in the reference list only contains subtitle of the paper. It should contain a main title.

Description in the Result section is not based on the results obtained. In lines 189-191, for example, ‘Figure 4 shows the results. As expected, LPS at the concentrations of 5 ng/mL and 50 ng/mL showed enhanced root development (Figure 4 a-d) and thin root generation (Fig. 4e)’ is described. On the other hand, figure 4 shows significant differences only in root length (a), surface area (in 5 pg/mL) (b), and forks (d). Such inaccurate notations are also found in explanations for the results of figure 2 (especially for results using Tsuru 2) and figure 3 (in surface area and between tips and forks). The effectiveness of bacterial cells and LPS should be discussed based on the accurate evaluation for results.

In line 230, ‘Since the R. sp. Tsuru2 and R. sp. Tsuru3 were not pure PNSB cultures’ is described. I could not understand this sentence. If Tsuru2 and Tsuru3 are not pure cultures of bacterium, how did the authors identify these strains as Rhodopseudomonas spp? The authors needs to clarify this issue.

Minor poins

Figure S1 is too small to understand the content and to read sentences and notations. So, enlarge the size of figure.

‘R. sp. Tsuru2’ and ‘R. sp. Tsuru3’ are irregular notations. I recommend to use only the strain names. Abbreviate a genus name in case of the binomial nomenclature including binomial species name.

In line 51, replace ‘treat’ with ‘treats’.

Italicize ‘Oryza sativa’.

Correct the unit for photon flux density as ‘micromol’ instead of ‘microM’.

In line 86, replace ‘use’ with ‘uses’.

Author Response

Thank you for your time and effort to review our manuscript and your quite helpful comments and suggestions to improve our manuscript. We revised our manuscript in accordance with your comments as follows.

(Comment) A method to prepare LPS from R. sphaeroides NBRC12203 should be included in the manuscript. In addition, describe how to confirm the identity and purity of LPS in the fraction and how to measure the quantity of LPS.

 (Answer) LPS from R. sphaeroides NBRC 12203 was purchased from InvivoGen (San Diego, USA). In the original manuscript, we described “LPS of R. sphaeroides ATCC 17023 was purchased from InvivoGen (San Diego, USA). (line 84 in the original manuscript)”. R. sphaeroides NBRC 12203 and R. sphaeroides ATCC 17023 are the identical strains with different culture collection numbers. We obtained R. sphaeroides NBRC 12203 from NBRC (Biological Resource Center, NITE, Japan), and call this strain as R. sphaeroides NBRC 12203 in the manuscript, while the manufacturer of LPS (InvivoGen) purified LPS from R. sphaeroides ATCC 17023. We think it may be better to call this LPS as LPS from R. sphaeroides NBRC 12203 to avoid confusion, and changed the description as follows (lines 97 – 101 in the revised manuscript)

2.3. LPS from Rhodobacter sphaeroides NBRC 12203 T

LPS of R. sphaeroides ATCC 17023 [21] was purchased from InvivoGen (San Diego, USA). R. sphaeroides NBRC 12203T and R. sphaeroidesATCC 17023 are the identical strains with different culture collection numbers. To avoid confusion, in the present paper we call this LPS as LPS from R. sphaeroides NBRC 12203T.

(Comment) The cultivation method of purple nonsulfur bacteria (Tsuru2, Tsuru3, and NBRC12203) is not described in the manuscript. Even though readers would look at reference 17 to know a method used for cultivation of Tsuru2 and Tsuru3, ref 17 includes a method using 24 well plate. I think a larger culture volume might be required for the rice seed soaking. The information for ref 17 in the reference list only contains subtitle of the paper. It should contain a main title.

(Answer) As suggested by the reviewer, we described the culture methods for strains Tsuru2 and Tsuru3. Please see lines 86 to 92 of the revised manuscript.

We checked the title of reference [17] and its full title is ‘Value-added recycling of distillation remnants of Kuma Shochu: A local traditional Japanese spirit, with photosynthetic bacteria’.

(Comment) Description in the Result section is not based on the results obtained. In lines 189-191, for example, ‘Figure 4 shows the results. As expected, LPS at the concentrations of 5 ng/mL and 50 ng/mL showed enhanced root development (Figure 4 a-d) and thin root generation (Fig. 4e)’ is described. On the other hand, figure 4 shows significant differences only in root length (a), surface area (in 5 pg/mL) (b), and forks (d). Such inaccurate notations are also found in explanations for the results of figure 2 (especially for results using Tsuru 2) and figure 3 (in surface area and between tips and forks). The effectiveness of bacterial cells and LPS should be discussed based on the accurate evaluation for results.

(Answer) Thank you for your suggestion. We revised the manuscript as suggested by the reviewer as follows.

For Figure 2, please see lines 169 to 174 in the revised manuscript.

For Figure 3, please see lines 204 to 212 in the revised manuscript.

For Figure 4, please see lines 227 to 230 in the revised manuscript.

(Comment) In line 230, ‘Since the R. sp. Tsuru2 and R. sp. Tsuru3 were not pure PNSB cultures’ is described. I could not understand this sentence. If Tsuru2 and Tsuru3 are not pure cultures of bacterium, how did the authors identify these strains as Rhodopseudomonas spp? The authors needs to clarify this issue.

(Answer) The authors are sorry for insufficient explanation of Rhodopseudomonas sp. Tsuru2 and Rhodobacter sp. Tsuru3. The identification of these strains was done by amplifying a part of pufL and puf M genes (operon) by PCR. These genes code one of the light reaction center proteins of photosynthetic bacteria, and heterotrophic (non-photosynthetic) bacteria do not have these genes. We identified the Rhodopseudomonas sp. Tsuru2 and Rhodobacter sp. Tsuru3 based on the nucleotide sequences of pufL and pufM genes. To clarify this issue, we added some explanation and two references in the revised manuscript (please see lines 74 – 83, and references [18] and [19]).

Minor points

(Comment) Figure S1 is too small to understand the content and to read sentences and notations. So, enlarge the size of figure.

(Answer) Thank you for kind suggestion. We enlarged the size of figure S1.

(Comment) ‘R. sp. Tsuru2’ and ‘R. sp. Tsuru3’ are irregular notations. I recommend to use only the strain names. Abbreviate a genus name in case of the binomial nomenclature including binomial species name.

(Answer) Thank you for your suggestion. We changed R. sp. Tsuru2 and R. sp. Tsuru3 to strain Tsuru2 and strain Tsuru3, respectively.

(Comment) In line 51, replace ‘treat’ with ‘treats’.

(Answer) We corrected ‘treat’ to ‘treats’ in the revised manuscript (line 53 in the revised manuscript).

(Comment) Italicize ‘Oryza sativa’.

(Answer) We Italicized ‘Oryza sativa’ (line 66 in the revised manuscript).

(Comment) Correct the unit for photon flux density as ‘micromol’ instead of ‘microM’.

(Answer) We corrected the unit for photon flux density as ‘micromol’ (line 92 in the revised manuscript).

(Comment) In line 86, replace ‘use’ with ‘uses’.

(Answer) We corrected ‘use’ to ‘uses’ (line 105 in the revised manuscript).

Reviewer 2 Report

The authors investigated the effects of seed bio-priming by purple non-sulfur bacteria (PNSB) on root development of rice. Actually, the work is very interesting, relevant and may have both fundamental and practical interests.

However, there are many confusing and incomprehensible points in this article. Please, look all my comments in the attached file with the paper.

Perhaps, the authors incompletely presented and described the data, or the setting of the experiments is incorrect. So, it is required to seriously revise the article, in particular, taking into account my comments indicated in the attached file.

Best wishes

Author Response

Dear Reviewer,

Thank you for your time and effort to review our manuscript and your quite helpful comments and suggestions to improve our manuscript. We revised our manuscript in accordance with your comments. For our answers to your comments, please see the attached PDF file.

Best regards,

Hitoshi Miyasaka

Round 2

Reviewer 1 Report

I think that the manuscript has been revised successfully and would contribute to broadening the applicability of PNSB in biotechnological fields.

Reviewer 2 Report

The authors made necessary corrections. The paper may be accepted for publication.

Best regards